# Impact of Uptake Period on ^18^F-DCFPyL-PSMA PET/CT Maximum Standardised Uptake Value

**DOI:** 10.3390/cancers17060960

**Published:** 2025-03-12

**Authors:** Anthony-Joe Nassour, Anika Jain, Hadia Khanani, Nicholas Hui, Nadine J. Thompson, Brian Sorensen, Sris Baskaranathan, Philip Bergersen, Venu Chalasani, Thomas Dean, Max Dias, Michael Wines, James Symons, Lisa Tarlinton, Henry Woo

**Affiliations:** 1Department of Urology, Sydney Adventist Hospital, Wahroonga, NSW 2076, Australia; 2SAN Nuclear Medicine and Radiology, Sydney Adventist Hospital, Wahroonga, NSW 2076, Australia; 3SAN Prostate Centre of Excellence, Sydney Adventist Hospital, Wahroonga, NSW 2076, Australia; 4iMED Radiology, Sydney, NSW 2046, Australia; 5Department of Urology, Blacktown and Mount Druitt Hospital, Blacktown, NSW 2148, Australia; 6Blacktown Mount Druitt Clinical School, Western Sydney University, Blacktown, NSW 2148, Australia

**Keywords:** prostate cancer, PSMA PET/CT, standardised uptake value, uptake time, uptake period

## Abstract

The accuracy of PSMA PET imaging can be influenced by the time between the injection of the imaging agent and when the scan is taken. This study explored how different time intervals (60, 90 and 120 min) affect the measurement of SUV_max_, a key value used in diagnosing prostate cancer. The results showed that longer times consistently led to higher SUV_max_ values, with the most significant change seen between 60 and 120 min. These findings suggest that waiting 120 min before scanning may provide the most accurate results for this type of imaging. Additional research is needed to confirm if this timing is effective for other similar imaging methods.

## 1. Introduction

Prostate cancer is the most prevalent cancer diagnosed in Australian men and is estimated to affect one in six by the age of 85 [1]. The advent of positron emission tomography/computed tomography (PET/CT) utilising a prostate-specific membrane antigen (PSMA) radioligand has marked a leap in diagnostic precision, significantly impacting decision-making processes and treatment algorithms compared to conventional computed tomography and bone scintigraphy [2,3]. PSMA PET/CT has cemented its place as the standard of care for primary staging and restaging in the setting of prostate specific antigen (PSA) biochemical recurrence, guiding a new era in the management of prostate cancer [4].

Gallium-68 labelled PSMA (68 Ga-PSMA) remains the most extensively studied radiotracer [5,6]. Despite widespread adoption, logistical constraints persist due to its production capacity, nuclear decay properties and increasing upfront cost for the generator and non-consumables [7]. 18 F-DCFPyL-PSMA is a second-generation fluorine-labelled radiotracer that provides comparable detection rates to 68 Ga-PSMA and inherently slightly higher resolution [8]. The favourable synthesis and decay properties of 18-F over 68-Ga permit the production of six, compared to four, batches from a single preparation [8]. In addition, the longer half-life of 110 min allows for centralised, larger-scale production with potential to overcome real-world logistical limitations [8]. This represents a more cost-effective option for low volume PSMA PET/CT centres and for countries where significant upfront investments are prohibitive.

The SUV_max_ represents the peak activity concentration of radiotracer uptake in a defined region of interest (ROI) and is a widely reported parameter in PSMA PET/CT imaging. Its magnitude has been shown to correlate with the detection and grade of clinically significant prostate cancer (csPCa) [9]. Emmett et al. introduced a five-point scoring system (PRIMARY score), evaluating PSMA patterns and intensity within the prostate. This study demonstrated that lesions with a SUV_max_ ≥ 12 were strongly predictive of clinically significant prostate cancer, with a 100% likelihood of csPCa for PRIMARY Score 5 [9].

However, while SUV_max_ serves as a valuable semi-quantitative biomarker, it is influenced by multiple factors, which can challenge its reproducibility across different imaging studies and centres. The injected radiotracer dose, the patient’s body weight (lean vs. gross) and the method of ROI delineation directly influence the SUV_max_ calculation and absolute value [10,11,12]. Without rigorous protocol standardisation, even minor variations in these factors can lead to significant cumulative variability in SUV_max_. A multicentre study demonstrated substantial differences in standardised uptake values (SUVs), including SUV_max_, across imaging centres when default PET protocols were applied, with measurement biases reaching up to 44% [12]. However, the implementation of standardised PET protocols reduced this variability, improving SUV accuracy to within 10%, highlighting the critical role of protocol harmonisation in ensuring reliable and consistent SUV quantification in preclinical PET/CT imaging [12].

This prospective study seeks to evaluate the impact of the uptake period (time between radiotracer injection and imaging) on the SUV_max_ using ^18^F-DCFPyL in PSMA PET/CT in the primary diagnosis of localised clinically significant prostate cancer.

## 2. Materials and Methods

### 2.1. Participants and Study Design

Sixty biopsy-naïve men with one or more PI-RADS 4 or 5 lesions of at least 10 mm on axial T2 multiparametric MRI were enrolled to undergo ^18^F-DCFPyL-PSMA PET/CT between January 2022 and December 2022 from a single Australian tertiary centre. The SUV_max_ was prospectively measured following an uptake period of 60, 90 and 120 min. Concordance with biopsy results or final histopathology was recorded. This patient cohort was derived from an HREC approved study published by Woo et al. investigating whether PET/MRI fusion could negate the need to biopsy prior to prostatectomy in a selected population of men [13]. Ethical approval encompassed the evaluation of PSMA PET/CT imaging as an independent modality within the diagnostic pathway for localised prostate cancer.

### 2.2. Ethical Considerations

The study protocol was reviewed and approved by the Adventist Healthcare Human Research Ethics Committee (HREC) (approval number 2018-042). All subjects provided informed consent prior to study participation.

### 2.3. PSMA PET/CT Protocol

Up to 350 MBq of ^18^F-DCFPyL was administered according to body weight as a slow bolus injection over 30 s. Scan time was 60, 90 and 120 min after injection from the vertex to the thighs with a noncontrast-enhanced low-dose CT scan after tracer injection using the following CT parameters: 3.75 mm slice thickness with PET attenuation correction reconstruction and standard reconstruction kernels; 120 keV and 80–200 mA (autoadjusted to minimise dose per patient body habitus); pitch of 0.984; large body field of view (FOV); helical rotation at 0.5 s per rotation; and a 512 matrix. The PET acquisition parameters were as follows: 3 min per bed position using a static acquisition and a 128 matrix, with scanning commencing at the pelvis and reconstructed via a measured attenuation correction method using a standard filter with two iterations and 24 subsets. Diagnostic contrast CT scans of the chest, abdomen and pelvis were performed as part of a usual standard-of-care examination using the following CT parameters: 1.25 mm slice thickness with soft reconstruction kernel; 120 keV and 100–800 mA (autoadjusted to minimise dose per patient body habitus); pitch of 0·516; large body FOV; helical rotation at 0.5 s per rotation at 1 mm intervals; and a 512 matrix. Intravenous contrast was administered at 1 mL/kg.

### 2.4. SUV_max_ Calculation

The SUV_max_ was calculated by defining the entirety of the tumour and using a volumetric region of interest. Accurate measurements, particularly with larger, heterogenous tumours with variable PSMA expression were achieved due to our approach using PSMA PET/MRI from the parent study [13]. SUV_max_ values were calculated in the standard fashion via GE AW PET software (version 4.6) and the technique was consistent across the entire cohort.

### 2.5. Statistical Analysis

Data analysis was conducted using R statistical software, version 4.1.0 (R Core Team 2021). Changes in SUV_max_ values were evaluated using the Wilcoxon signed-rank test. Univariate regression analysis with generalised estimating equations was used to adjust for interlesion correlation within individual participants. The optimal SUV_max_ cutoff value following an uptake period of 90 min was obtained by constructing a receiver operating characteristic (ROC) curve and employing the Youden index. Linear regression analysis was used to determine the relationship between the SUV_max_ and ISUP grade group (GG); *p*-values < 0.05 were considered statistically significant.

## 3. Results

### 3.1. Participant and Lesion Characteristics

A total of 69 lesions were biopsied from 60 participants. A total of 35% (24/69) of these lesions were PI-RADS 4 and 65% (45/69) were PI-RADS 5. Participant median age was 68 years with an interquartile range (IQR) of 11. The medians for PSA, PSA density, prostate volume, and SUV_max_ were 4.3 ng/mL (IQR 4), 0.15 ng/mL/cc (IQR 0.08), 40 cc (IQR 24), and 11 (IQR 22.8), respectively.

### 3.2. Impact of Uptake Period on SUV_max_

Mean absolute differences in the SUV_max_ at 60 vs. 90, 60 vs. 120, and 90 vs. 120 min uptake periods were 3.23 (SD 4.76), 4.53 (SD 7.33), and 3.24 (SD 4.56), respectively (Table 1). This represents a statistically significant systematic increase in the SUV_max_ (*p* < 0.001) with the increasing uptake period. The interval between uptake periods of 60 vs. 120 min represented the largest SUV_max_ change of 29.98%. A visual representation of increasing intensity with an increasing uptake period from 90 to 120 min from the same subject can be appreciated in Figure 1. The respective SUV_max_ for this representative subject at 60, 90 and 120 min were 7.51, 9.39 and 15.27 respectively. Unfortunately, PSMA PET/CT fusion images at 60 min were lost to archive.

### 3.3. Correlation of SUV_max_ and Grade Group

The SUV_max_ increased by 6.06 (95% CI 3.54 to 8.58) for every unit increase in the GG (*p* < 0.0001). However, correlation between the change in SUV_max_ and GG was only found to be significant for GG-2 lesions (*p* < 0.05) at uptake periods of 120 min compared to 60 min (Table 2).

### 3.4. SUV_max_ Thresholds and Clinical Implications

When correlating the SUV_max_ to a prostate biopsy or final histopathology, clinically significant prostate cancer (CSPCa) was exclusively detected when the SUV_max_ exceeded 9.1, 5.3 and 9.5 for the 60, 90 and 120 min uptake periods, respectively (Table 3). The optimal SUV_max_ threshold with the highest Youden index (0.49) was 10.73 (sensitivity 56.9%, specificity 91.7%, PPV 57.9% and NPV 91.4%). At this threshold, twenty-five (43.1%) clinically significant prostate cancers would have been missed (*p* < 0.05). By contrast, using a lower threshold of 5.34 (at an uptake period of 90 min) detected an increasing number of clinically insignificant prostate cancer (Grade Group 1) at the cost of omitting clinically significant prostate cancer (sensitivity 75.9%, specificity 66.7%, PPV 31.4% and NPV 93.3%) which may ultimately affect treatment pathways.

## 4. Discussion

SUV_max_ is a semi-quantitative measure widely utilised in PSMA PET/CT reporting to differentiate benign from malignant tissue [14]. Its popularity is primarily related to its low observer-dependence, simplicity and reproducibility [14]. However, the SUV_max_ can be influenced by various tracer, physiological, pathological and technical factors [15].

This prospective study underscores the importance of considering uptake period when interpreting the SUV_max_ for the primary diagnosis of localised prostate cancer in biopsy-naïve men following administration of ^18^F-DCFPyL-PSMA radiotracer. We demonstrated a statistically significant increase in SUV_max_ with a rising uptake period. The greatest difference of ~30% occurred with an increase in the uptake period from 60 to 120 min. Our findings are in keeping with existing retrospective data by Rowe et al. and Wondergem et al., who both reported increased radiotracer uptake using ^18^F-DCFPyL radiotracer at 120 min compared to 60 min [3,16]. Similar findings were also observed from more recent studies by Tian et al. [14]. Although all three studies, Rowe et al., Wondergem et al. and Tian et al., demonstrate an increasing SUV_max_ with an increased uptake period, it is crucial to recognise the significant differences in their population cohorts and study focus. Our study, with a prospective design, included 60 biopsy-naïve patients with localised prostate cancer and MRI-detectable lesions, making it highly relevant for primary diagnosis. In contrast, Rowe et al. had a small cohort of only nine patients, all in the context of biochemical recurrence. Tian et al. included 38 patients, but only 12 were undergoing primary staging, with the study’s primary focus on differential diagnosis between benign and malignant lesions rather than SUV_max_ kinetics. While Wondergem et al. had a slightly larger cohort of 66 patients, only 21 were for primary staging, with the remainder undergoing imaging for biochemical recurrence or metastatic disease. Although their study partially analysed SUV_max_ in a primary diagnostic setting, our research is superior due to its prospective nature, larger sample size and quantitative statistical analysis of SUV_max_ changes, providing stronger evidence to guide clinical decision-making.

Theoretically, Fluorine-18 based radiotracers are expected to provide a superior tumour/background ratio and enhanced spatial resolution compared to Gallium-68 tracers due to a prolonged half-life, higher positron yield, shorter positron range and lower positron energy [17]. However, head-to-head comparisons between ^18^F-DCFPyL and ^68^Ga-PSMA-11 remain limited. A retrospective study assessing biodistribution of ^68^Ga-PSMA-11 and ^18^F-DCFPyL in 43 men who underwent scans with both tracers reported similar biodistribution and urinary excretion [5]. Although bladder SUV_max_ was higher in the ^18^F-DCFPyL group the longer half-life of ^18^F allows for urinary dilution and delayed post-void imaging [5].

Early prospective studies involving 14 participants who underwent PSMA PET/CT with both agents reported that lesion detection with ^18^F-DCFPyL was noninferior, with a notably higher SUV_max_ for the same lesion [8]. In contrast, more recent retrospective studies evaluating intraprostatic lesions prior to robotic radical prostatectomy did not demonstrate a difference in the SUV_max_ between ^68^Ga-PSMA-11 or ^18^F-DCFPyL [18]. The median SUV_max_ for the dominant intraprostatic lesion for those who underwent a ^68^Ga-PSMA-11-PET/CT and ^18^F-PSMA-PET/CT scan was 8.1 (4.9–14.5) and 7.8 (5.8–13.8), respectively. It is worth noting that the acquisition period for the ^18^F-PSMA-PET/CT was at 118 min post tracer injection [18].

Our study highlights the dynamic nature of SUV_max_ in relation to uptake period. The data presented supports imaging at 120 min post injection of ^18^F-DCFPyL radiotracer. This is consistent with data from other prominent, high volume PSMA PET/CT centres that have since modified their protocols to image at 120 min when utilising ^18^F-based tracers, reflecting a shift in best practices [5,17]. Existing guidelines continue to recommend an uptake period of 60 min for most tracers, including ^68^Ga-PSMA and ^18^F-DCFPyL [4]. However, based on our findings, we recommend an uptake period of at least 90 min when using ^18^F-DCFPyL for prostate cancer imaging, potentially improving lesion detectability and diagnostic confidence in clinical practice.

Key strengths of this study lie in its prospective design, stringent methodology and patient selection. These factors reduce heterogeneity and observer bias to produce reliable and reproducible results. Limitations of this study include single-centre selection bias and a relatively small sample size, which may impact the generalisability of our findings. Moreover, we did not compare SUV_max_ values between benign and malignant lesions, nor did we consider uptake periods longer than 120 min and it is unclear whether longer uptake periods might further improve sensitivity and specificity of ^18^F-DCFPyl-PSMA PET/CT. Our findings support earlier results reported by Wondergem et al., who demonstrated that 18F-DCFPyL PET/CT at 60 and 120 min improves detection rates and image quality while providing insight into activity kinetics and biodistribution [16]. The applicability of our findings to other imaging protocols and tracers is unclear.

In conclusion, this prospective study enriches the existing literature on PSMA PET/CT imaging and SUV_max_ variability, supporting the findings of retrospective studies and highlighting the need for standardised imaging protocols based on the type of radiotracer used to improve diagnostic accuracy and reproducibility.

## 5. Conclusions

SUV_max_ is a dynamic variable significantly affected by uptake period. Our study supports image acquisition at 120 min following injection of ^18^F-DCFPyL radiotracer and underscores the need for PSMA PET/CT protocol standardisation to generate reproducible results and reliable interpretation. Further studies are needed to determine if this acquisition period is applicable to other ^18^F-based PSMA radiotracers.

## Figures and Tables

**Figure 1 cancers-17-00960-f001:**
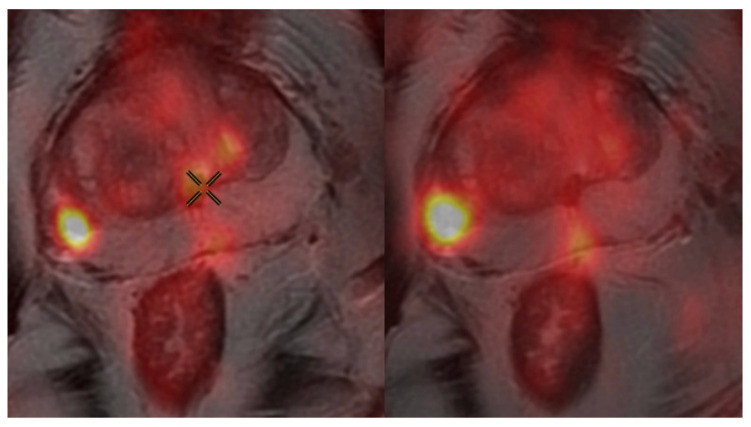
Image of PSMA PET/CT fusion from the same patient demonstrating progressive enhancement of avid right peripheral zone intraprostatic lesion taken at 90 min (**left**) and 120 min (**right**) with SUV_max_ of 9.39 and 15.27 respectively.

**Table 1 cancers-17-00960-t001:** Mean absolute and percent differences of SUV_max_ at 60, 90 and 120 min uptake periods (*n* = 69).

Uptake Period (min)	Mean Absolute Difference Between SUV_max_ (SD)	Mean % Difference Between SUV_max_ (SD)	Significance
60 vs. 90	3.23 (4.76)	21.97%	*p* < 0.001
90 vs. 120	3.24 (4.56)	10.35%	*p* < 0.001
60 vs. 120	4.53 (7.33)	29.98%	*p* < 0.001

**Table 2 cancers-17-00960-t002:** Grade group and associated SUV_max_.

Grade Group	Number	SUV_max_
No cancer	11	6.69
1	1	5.30
2	24	8.12
3	18	23.7
4	5	39.4
5	11	29.5

**Table 3 cancers-17-00960-t003:** List of predictive value at various SUV_max_ cut-off levels.

SUV_max_ Cut-Off Value	Sensitivity	Specificity	PPV	NPV	Youden Index
10.73	56.9%	91.7%	57.9%	91.4%	0.49
5.34	75.9%	66.7%	31.4%	93.3%	0.43
3.41	94.8%	50.0%	27.5%	98.0%	0.45

## Data Availability

Data are contained within this article.

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
