# Peer review of "Impact of Uptake Period on 18F-DCFPyL-PSMA PET/CT Maximum Standardised Uptake Value"

_cancers, 2025, doi:10.3390/cancers17060960_

Round 1
Reviewer 1 Report
Comments and Suggestions for Authors
This research examines the influence of uptake time on SUVmax values in 18F-DCFPyL-PSMA PET/CT imaging for identifying clinically relevant prostate cancer. The research is methodically organized, featuring a prospective design. A minor revision is recommended .
The real PET images illustrating lesion visibility at various uptake intervals to visually corroborate the study's conclusions would be appreciated. And that is supposed to include:
- A single patient’s PSMA PET/CT scan at 60, 90, and 120 minutes, demonstrating incremental SUVmax variations.
- A case in which a lesion was poorly seen before 60 minutes but grew more distinct at 120 minutes.
- A comparison of benign and malignant lesions at 120 minutes to reinforce SUVmax thresholds.
- Extra assessment of uptakes durations over 120 minutes.
Author Response
We thank the reviewer for their responses. We certainly have tried our best to address their concerns in the best way possible and with as much detail as necessary.
We hope they will be satisfied with our responses.
Kind regards,
Professor H. Woo

Reviewer 2 Report
Comments and Suggestions for Authors
The article by Nassour et al. entitled “Impact of Uptake Period on 18F-DCFPyL-PSMA PET/CT maximum standardized uptake value” evaluates how different time-point affect the measurement of SUVmax in 60 biopsy-naive prostate cancer patients who underwent 18F-DCFPyL-PSMA PET/CT.Overall, the article achieves its objective by drawing conclusions that are already known from larger cohorts. Some comments are provided below:
Introduction:
- The introduction lacks details on the population studied, so that the role of PSMA PET in primary tumour diagnosis could be more detailed and realistic. Overstatement should be avoided (e.g. 'Landamark study....').
- Reference 4,5 not appropriate. Refer to oncology guidelines and/or EANM guideline.
- Lack of adequate introduction to radiopharmaceutical investigated, highlighting why this study is needed.
Materials and methods
- The cohort is homogeneous but relatively small considering the pathology studied.
- Statistical analysis: Why was the cut-off value calculated at 90 minutes? Also considering that the conclusions of the manuscript are drawn at 120 minutes.
- How was SUVmax calculated? was a VOI used? Please define
- Why was it not considered to also calculate a ratio (e.g. lesion to normal ratio) with a healthy prostate or other background? to normalise SUVmax values.
Results:
- Table 1 is just a repetition of the text, consider removing it.
- According to guidelines, fluorinates are generally acquired at 90 minutes post-injection, the study indeed confirms that between 90 and 120 minutes the % difference in SUVmax is only 10%. This is in line with current clinical practice.
- Tables 3 and 4 are reversed.
- Paragraph “SUVmax thresholds and clinical implications”: I cannot understand why certain thresholds (9.1, 5.3 and 9.5) are reported at different time points (60, 90, 120 minutes). Others are discussed later and in the table. Please clarify this point in the text and improve the table to make it clearer. Consider including ROC curves.
Discussion:
- Significantly improve the discussion by highlighting the innovative and different points compared to the already mentioned studies by Rowe et al. and Wondergem et al.
- In addition, when comparing the two radiopharmaceuticals, please highlight the differences in the reported SUVmax acquisition times.
- Page 5, lines 184-185: Please add a reference. Not convinced.
Minor comment:
- Add one or two figures as case examples.
- Correct some minor spelling errors throughout the text.
Author Response

(The authors gave the same response as above.)
